# Understanding the Impact of Chronic Non-Cancer Pain on Daily Life from a Gender Perspective Using the PAIN_Integral Scale©

**DOI:** 10.3390/healthcare12060695

**Published:** 2024-03-20

**Authors:** Rocío Cáceres-Matos, Soledad Vázquez-Santiago, Manuel Pabón-Carrasco, Andrés Cabrera-León, Eugenia Gil-García

**Affiliations:** 1Research Group CTS-1050: “Complex Care, Chronicity and Health Outcomes”, Nursing Department, Faculty of Nursing, Physiotherapy and Podiatry, University of Seville, 41009 Seville, Spain; rcaceres3@us.es (R.C.-M.); egil@us.es (E.G.-G.); 2Department of Public Health, Andalusian School of Public Health, 18011 Granada, Spain; andres.cabrera.easp@juntadeandalucia.es; 3Biomedical Research Consortium in Epidemiology and Public Health Network (CIBERESP), 28029 Madrid, Spain

**Keywords:** chronic pain, factor analysis, statistical, female, male, surveys and questionnaires, validation

## Abstract

The experience of chronic non-cancer pain differs between women and men due to gender-related factors. This study (1) assessed the difference in responses to the impact of chronic non-cancer pain on daily life in women and men using the PAIN_Integral Scale© and (2) evaluated its invariance through multigroup confirmatory factor analysis. This was conducted by means of an analysis of invariance through a multigroup confirmatory factor analysis. A cross-sectional sample of 400 participants over 18 years of age with Chronic Non-Oncological Pain in Pain Units and Primary Care Centres belonging to the Spanish Public Health System was recruited (January to March 2020). An analysis was performed to assess whether any of the items in the instrument showed different behaviours. All analyses were performed using AMOS^®^ v.26 software. The results showed that the structure of the PAIN_Integral© Scale remained adequate when analysing its invariance in women and men, showing no metric, scalar and/or strict invariance. Therefore, these results indicated that the PAIN_Integral Scale© instrument has a different interpretation for women and men, identifying eight items with a singular functioning in both sexes and belonging to the subscales of proactivity, resilience and support network. These findings can be explained by gender stereotypes, since the dimensions where there are differences have an important social burden.

## 1. Introduction

Chronic Pain (CP) is considered a type of pain that persists continuously or intermittently for more than three months [1], and when not associated with a neoplastic condition, it is denoted as Chronic Non-Cancer Pain (CNCP) [2]. Epidemiological studies calculate the prevalence of CP among 20% of the world population [3].

Consensus exists that a correct biopsychosocial assessment centred on the person improves the intensity of CNCP [4] but fundamentally it impacts daily life [4,5]. Similarly, assessment is crucial for determining the most suitable intervention or treatment and forms the basis upon which the efficacy of its methodology hinges [6]. 

In this sense, numerous investigations have studied the impact of CNCP on daily life. They established that CNCP influences aspects such as sleep quality [7] and limitations on daily functioning [8], but also, psychological factors such as anxiety, depression [9], self-esteem [10], pain coping [11] and resilience [8]. In addition, the experience of CNCP is uneven in different societies given the connotations that it has in each culture [12,13] and social aspects [14], as well as by sex and gender [15]. 

According to sex, studies suggest that women and men exhibit distinct pain responses, which appear to be attributed to genetic, anatomical, physiological, hormonal and neural factors [16]. 

Regarding gender, disparities may arise from the different roles and behaviours expected of men and women; that is, gender stereotypes result from the interaction of sex with differential gender socialisations in the experience of CNCP throughout life [17]. In this regard, gender stereotypes also contribute to variations in exposure to healthcare outcomes and treatments, making it an aspect that should be considered in clinical practice [18,19]. 

Over the years, numerous instruments have been developed and validated to assess the different areas on which CNCP has an impact. However, the use of these instruments remains a challenge and the data indicate that only 10% of healthcare professionals use them [20,21]. The low use by healthcare professionals can be justified by the large number of instruments to assess each of the aspects affected by CNCP, the variability of the methodologies and the lack of knowledge healthcare professionals have about them [22,23]. Considering these reasons, the PAIN_Integral Scale© instrument was designed and preliminarily validated in response to the need to combine all the biological, psychological and sociocultural aspects that are altered by CNCP in a single instrument, speeding up the assessment process [24,25]. 

Similarly, there are precedents where differences between women and men were evidenced in scales validated in the CNCP population and which are widely known and used in clinical practice, such as the Pain Catastrophising Pain Scale [26] or the McGill Short Form Pain Questionnaire [27]. Therefore, in order to verify whether there are differences in the meaning of the items in the PAIN_Integral Scale© instrument between women and men, a Multigroup Confirmatory Factor Analysis (MGCFA) of invariance has been carried out. 

MGCFA technique aims to compare groups and determine whether a construct has the same meaning for different groups or samples due to its idiosyncrasies and characteristics. Thus, an invariant model ensures individuals of different groups are interpreting the items of the questionnaire and their meanings similarly, regardless of group membership (e.g., sex: women and men). This confirms that scores from the instrument truly correspond with the underlying constructs and are not due to group-specific attributions [28]. The objectives of this study were to test the structure of the PAIN_Integral Scale© using MGCFA to assess its invariance in women and men and to analyse the items that show a differential functioning between both groups.

## 2. Materials and Methods

### 2.1. Design

The study comprised a cross-sectional multicentre design following the recommendations from the guide Sex and Gender Equity in Research (SAGER), which was designed to guide authors in the communication of research results for sex–gender [29]. The STROBE checklist was also adhered to (see Appendix A).

### 2.2. Study Setting and Sampling

The sample size was designed following the recommendations of McCallum [30] and the COSMIN Checklist Statement [31], which estimates a minimum of five and seven individuals per item, respectively, which means a range from 180 to 252 participants. The minimum sample size to be used was also calculated using G*Power V.3.1.9.4 for a large effective size (f^2^ = 0.35), a power of 0.90 and a statistical significance of 95% (α = 0.05) [32,33,34]. The minimum number of subjects was considered to be 386, but the sample size of our study was larger with a total of 400 subjects.

### 2.3. Inclusion and Exclusion Criteria

Inclusion criteria were being over 18 years of age and suffering from any CNCP condition. Exclusion criteria covered patients who suffered from cancer pain, neurodegenerative diseases, cognitive impairment or difficulties with oral communication in the Spanish language.

### 2.4. Instrument with Validity and Reliability

The impact of CNCP on daily life was assessed using the PAIN_Integral Scale©. This instrument facilitates the consolidation of various areas affected by CNCP into a unified scale, expediting the assessment process and considering aspects not addressed by other instruments. The PAIN_Integral Scale© is an instrument composed of 36 items with a Type-Likert scale from one to five points available in the Spanish language. The results of its validation study showed adequate reliability (α = 0.72) and a structure of nine dimensions that explained 68.22% of the variance. The nine dimensions are self-care, mobility, sleep, treatment compliance, proactivity, resilience, support network, hopelessness due to pain and pain catastrophising. The results of the Confirmatory Factor Analysis (CFA) demonstrated a good fit for the proposed 9-subscale model Goodness of Fit Index (GFI = 0.93), the Standardised Root Mean Residual (SRMR = 0.05), and the Root Mean Square Error of Approximation (RMSEA = 0.06). Scores on the scale range from 36 to 180 points and are divided into three intervals (36–130: Severe impact; 131–135: Moderate impact; 136–180: Mild impact) [24,25]. 

### 2.5. Data Collection 

Data from participants were collected between January and March 2020 at Pain Units and Primary Healthcare Centres belonging to the Spanish Public Health System (Virgen del Rocio University Hospital; Virgen Macarena University Hospital; Virgen de Valme University Hospital and San Juan de Dios Aljarafe Hospital) in the province of Seville, in southern Spain. A trained researcher asked all participants every question so there were no missing items. The participants’ responses were entered in a Google form to create the database. Data on sociodemographic and clinical variables were collected, as presented in Table 1.

### 2.6. Data Analysis

Descriptive statistics were performed to summarise the characteristics of the sample. Continuous variables were expressed by means ($\bar{x}$) with the corresponding confidence intervals (CIs), and categorical variables were expressed as percentages (%) and CIs. The comparison of the different groups’ proportions was assessed using the Chi-Square test (χ^2^ test) for categorical variables and Student’s T-Test for continuous variables. Significance was considered to be 5% in all descriptive analyses using version 2.7.2 of the free software R© (the R project, Auckland, New Zealand).

Invariant analysis was performed by MGCFA and was conducted using an unconstrained–constrained approach. First, an unconstrained model was run, which allowed parameters to vary freely. The unconstrained model, also known as a configural model, means that the factors have identical items in both groups and serve as a baseline which subsequent models are tested against. This analysis was followed by a fully constrained model, where parameters were constrained to be equivalent across groups (sex: women/men), assessing metric and scalar invariance. Metric invariance assesses whether the factor loadings are equal across groups; nevertheless, metric invariance ensures that the meaning of the common factors is similar across groups. Scalar invariance ensures that the intercepts of the items are equal in the different groups, which means that they are not contaminated or influenced by external factors (group-specific attributes). Scalar invariance implies that the means of the latent variables are comparable across groups [35,36]. 

The structural invariance assumption is supported if MGCFA meets the following criteria [37]: (1)The model specifying the items measuring each latent variable fits the data well. Several fit indices were considered to test the CFA structure of the model: RMSEA < 0.08 [38,39]; Comparative Fit Index (CFI > 0.90) [39,40]; χ^2^ test (a non-significant χ^2^ test (*p*-value > 0.05) indicates an acceptable model structure); and the relationship between χ^2^ and the degrees of freedom (df) (χ^2^/df < 3) [38,41].(2)All factor loadings are substantial (above 0.30) and statistically significant.(3)No large modification indices exist that point to model miss-specifications. Model fit was compared using χ^2^ difference test (χ^2^_diff_), given the change in degrees of freedom between models with a cut-off of 0.01 [35,36,42]. Differences in CFI (ΔCFI) and RMSEA (ΔRMSEA) lower than 0.01 and 0.015, respectively, were also required between unconstrained and constrained models.

Structural analyses were performed using the Analysis of Moment Structures (AMOS©) version 26 software (IBM Corp., Armonk, NY, USA). 

To assess whether any of the items on the scale show a different relationship in the construct measured by the groups, the Item Differential Functioning (IDF) was evaluated. The Rasch method was used if the following assumptions were met: (1) scale invariance; (2) homogeneity of the items; and (3) local independence in the responses of the items. If these assumptions were not met, the χ^2^ test was used to examine the distribution of responses between the groups [43,44]. In this last case, the effective size was evaluated using the contingency coefficient Phi (Φ), considering a small effective size (Φ ≤ 30), medium (30 < Φ ≤ 50) and large (Φ > 50). These analyses were carried out using the Statistical Package for the Social Sciences (SPSS©) version 26 software (IBM Corp., Armonk, NY, USA). 

### 2.7. Ethical Considerations

The research committee of the Virgen Macarena-Virgen del Rocio University Hospital approved this study with the code 1373-N-20. All patients who met the inclusion criteria were informed about the study and invited to participate by providing written or verbal informed consent. Regarding this last one, since the study did not involve a modification of the treatment, progression or course of the disease, and it is an observational study, it could be considered a low risk for the participating individuals. The Research Ethics Committee approved this option of verbal consent, as outlined in the study protocol. Likewise, similar to those who provided written verbal consent, participants were given the participant information sheet, and all aspects related to the study were explained to them. Participants responded affirmatively and clearly expressed their interest in being part of the study.

## 3. Results

### 3.1. Characteristics of the Sample

This study’s subjects comprised a total of 252 women (63%) and 148 men (37%). Table 1 shows data on age, PAIN_Integral Scale© score, centre, marital status and employment situation, among others disaggregated by sex (Table 2).

### 3.2. Evaluation of the Invariance of the Measurement Model

Results from the MGCFA and χ^2^_diff_ test indicated that moderation on sex was present at the model level, which means that there were differences in the structure of the instrument between women and men. Firstly, the unconstrained model of the nine-solution model for women and men was assessed. Fit indices showed that the fit of the model data was adequate (CFI = 0.913; RMSEA = 0.038, 95% CI (0.034–0.041); χ^2^/df = 1.56) to compare the invariance with the metric and scalar models (Table 3). In addition, all standardised factor loadings for this nine-factor model were from 0.49 to 0.96 in women and men (Table 4). 

Table 3 shows the comparison of the fit indicators between the different calculated invariance models. Firstly, the metric invariance model was tested, in which the factor loadings were restricted to be equal between men and women. The indices showed that the model fitted well, but when compared with the unconstrained model, χ^2^_diff_ was significant (*p* < 0.001). However, the ΔRMSEA was less than 0.015 and ΔCFI was lower than 0.01. These results mean the model does not show metric invariance (Table 3). 

The test of the scalar invariance model, in which the intercepts, in addition to the factor loadings, were restricted to be equal between the groups (by sex), showed a good fit. When compared with the metric invariance model, significant changes were observed in χ^2^_diff_ (*p* < 0.001), while conversely, no significant changes were observed in the RMSEA (ΔRMSEA < 0.015) or in the CFI (ΔCFI < 0.01). These results mean the model does not show scalar invariance. 

Finally, in the strict invariance model, in which the error variances were restricted, in addition to factor loadings and intercepts, the structure showed a good fit. However, once it was compared with the scalar invariance model, significant changes were observed in χ^2^_diff_ (*p* < 0.001). Conversely, no significant changes were observed in the RMSEA (ΔRMSEA < 0.015) or the CFI (ΔCFI < 0.01). These results mean the model does not show strict invariance.

On the one hand, these results indicate that the structure of the instrument remained adequate when analysed separately for men and women. On the other hand, the overall understanding of the theoretical concept of the impact of CNCP on daily life is different for both sexes due to the characteristics of each group.

### 3.3. Evaluation of the Differential Functioning of the Item

The Item Differential Functioning was assessed using the χ^2^ test, since the data did not meet the assumptions for the Rasch method. Eight items were identified as items with different functions between women and men (items 16, 20, 23, 26, 27, 28, 29 and 30) (Table 5). These items belong to the subscales of proactivity, resilience and social support.

## 4. Discussion

The scientific literature recognises that there are differences in the experience of CNCP in women and men and that these are complex due to the cultural and social construction of gender [45]. Therefore, it is necessary to assess whether there are differences in the understanding of the items between both groups. The objectives of this study were to test the structure of the PAIN_Integral Scale© using MGCFA to assess its invariance in women and men and to analyse the items that show different functions between both of them.

The results are based on the fact that the fit indices of the original nine-factor, 36-item PAIN_Integral Scale© met the criteria for goodness of fit. Although previous studies did not find differences between women and men in the mean scores of the complete scale [24,25], when the invariance of the instrument was verified from the comparison of the metric (*p* < 0.001), scalar (*p* < 0.001) and strict invariance (*p* < 0.001) models, differences were identified between women and men.

However, the good fit indices of each model when the elements of the factorial structure remain invariant based on sex, except for one of the parameters of each model, means that partial invariance can be assumed [46] since strict invariance tests are excessively restrictive [47]. This also seems to indicate that the non-invariance between women and men is not due to an inadequate fit to the measurement or structural model. 

Therefore, the main results of this study show that differences in responses between men and women were detected, which may reflect differences in the interpretation of the questions or alternatively, differences in the impact of pain. The discernment of differences appears to be manifested in eight items of the PAIN_Integral Scale©, and consequently, in the constructs to which these items belong. On the other, these differences may be explained by gender reasons. Since there is no exact definition of the concept of the impact of the CNCP on daily life or other scales that measure this theoretical construct, it is not possible to compare the existence or non-existence of invariance in other instruments [24]. However, differences between women and men have also been found in scales validated in the population with CNCP and that are widely known and used in clinical practice, such as the Pain Catastrophising Scale [26] or the Short-Form McGill Pain Questionnaire [27].

With respect to the items, IDF analysis has shown that the non-invariance could be explained by eight items (item 16, item 20, item 23 and items 26 to 30). These items belong to proactivity, resilience and support network subscales. Item 16 “I try to know more about my pain so I can cope” belongs to the proactivity subscale and showed worse scores in women than in men. Proactivity is defined as “the processes which are employed to detect and prevent probable goal threats while working towards personal goals” [48]. 

In this sense, studies indicate that coping strategies in women and men are based on psychosocial aspects and are influenced by gendered expectations [49,50] and societal beliefs about how women and men are expected to behave [51]. While men use more distractive and problem-focused behaviours, women use a wider range of coping strategies, such as more emotionally focused strategies and social supports, which are often ineffective [11,46]. In line with the results for this item, a study carried out by Ouwehand et al. (2008) found that men used more proactive coping strategies than women [52]. 

Item 20 “I think I’m a strong person” belongs to the resilience construct and showed higher scores in women than in men. The conception of strength is determined and characterised by the social and cultural context, as well as by gender socialisation learned throughout life, known as a mechanical metaphor for pain. This means that the female body has been credited with the ability to withstand extreme pain throughout their lives [53]. The same occurs in dysmenorrhea, where women do not seek health care since they have normalised the pain [54]. In a qualitative study carried out by Chen et al. (2018), the participating women accepted dysmenorrhea as “something normal” or “something to live with” [55] (Chen et al., 2018). Therefore, this experience of constant pain throughout life is integrated into other painful experiences, such as CNCP [56].

On the other hand, while women are educated during childhood to verbalise discomfort and recognise the presence of the disease [18,51], for men, the Traditional Hegemonic Model of Masculinity (THMM) has determined a representation of masculine stoicism in the face of expressing physical and emotional discomfort that questions masculinity in its absence [53]. The male ideal based on this THMM positions men to be in control and fear vulnerability, strength and self-sufficiency [57], which is associated with worse health outcomes and resistance to seeking health care, among other aspects [57,58]. This produces a greater frequentation of health services by women, mistakenly understood as a greater morbidity instead of a greater predisposition to consult [59,60].

The remaining six items (item 23 “Do you have someone you can count on when you need to talk”; item 26—“Do you have someone to inform you and help you understand the situation”; item 27—“Do you have someone to do things with you to forget your problems”; item 28—“Do you have someone to help you with your households chores if you are sick”; item 29—“Do you have anyone to have fun with”; and item 30—“Do you have someone who understands your problems”) belong to the social support construct. Considering that a support network is defined as “information leading the subject to believe that he or she is cared for and loved, esteemed, and a member of a network of mutual obligation” [61,62], this concept has been related to a positive adaptation to pain [63].

All of these six items showed higher scores in men and these results are not consistent with those found in other studies related to support networks. According to Samulowitz et al. (2022), men showed lower emotional social support (OR = 0.54 (95 CI 0.39–0.74) [64]. Conversely, social and emotional skills have traditionally been associated with women in the Traditional Hegemonic Femininity Model, while the capacity for decision-making and self-confidence is related to the THMM [65,66]. Regarding the latter, men tend to have more instrumental support in the tasks and management of care and disease. However, in women, having the responsibility of domestic work, double and triple shifts and little support in care tasks and management have been associated with poorer health, especially in women who suffer from CNCP [67,68,69]. Other studies have also found that loneliness is more prevalent in women than in men and that these differences are accentuated when there is a disabling illness [70]. In fact, in other pathologies such as cardiovascular disease, it was found that in women, loneliness increased their suffering by 13% [71].

### 4.1. Strength and Limitations of the Work

As strengths, the results of this study have direct implications for clinical practice, from an economic and health perspective. CNCP generates significant costs for healthcare systems. The PAIN_Integral Scale© instrument makes it possible to identify the level of impact of CNCP at an early stage, favouring the approach and reducing the health cost, considering the differences between women and men. Regarding methodological aspects, a sample size much larger than the minimum required was used. In addition, the analysis of invariance using MGCFA has recently been included in the COSMIN Checklist Statement, which also justifies the need to carry out these analyses in psychometric validation studies [31].

As limitations, despite the differences found, we must take into account certain variables that can modify the response, mainly marital status and age. Marital status may be related to the participants’ support network. In addition, age is intimately related to the generational experiences where the individual develops and, therefore, this may influence gender roles. On the other hand, racial, ethnic and cultural differences have not been considered. In this regard, only a binary conception of the sex–gender system has been taken into account, not contemplating other identities or non-normative realities. On the other hand, the limitations derived from the study design itself (cross-sectional descriptive study), as well as the self-report data, must be taken into account.

### 4.2. Recommendations for Further Research

In future lines of research, racial, ethnic and cultural differences should be considered, besides contemplating other identities or non-normative realities. Therefore, since both the CNCP and gender differences are mediated by the aforementioned aspects, it would be necessary to replicate the study in the future, evaluating the understanding of the items from this intersectional perspective. An example of this is the analysis of invariance in widely known instruments such as the Pain Catastrophising Scale in Native American and Non-Hispanic White Adults [72] and the Pain Assessment in Advanced Dementia (PAINAD) between White and Black people [73]. Finally, much like the differences in item comprehension have been evaluated, it would be interesting, in future studies, to analyse the similarities found between women and men, making this another perspective line to consider. 

### 4.3. Implications for Policy and Practice

Gender inequalities across the world cause damage and this justifies comprehensive action in health at all levels [74]. According to Westergaard et al. (2019), women are diagnosed later than men in more than 700 diseases, including CNCP [75], in addition to there being a lack of care centred on women [76]. A traditional example of this is the management of cardiovascular diseases since they were identified in the 1930s [77,78,79]. According to authors such as Vlassof and García-Montero (2002), gender is key to understanding all dimensions of healthcare, and consequently, a transformation to integrate gender perspective is necessary [80]. Other authors highlight the need to develop practical tools that facilitate the application of interventions taking gender into account, allowing the treatment to be adapted to women who have traditionally been treated with treatments made for men [81].

Integrating sex and gender analysis into the design of research and as categories or variables is also important to avoid bias in the research [82]. This is supported by a whole series of recommendations and regulations at an international level, such as those of the European Commission (2019) [82] or the United States National Institutes of Health [83,84], as well as the Millennium Development Goals and Sustainable Development Goals of the United Nations [85].

Regarding the impact on clinical practices, in a study carried out by Stockbridge et al. (2015) in the United States, authors found that health spending per person increased by 4000 USD per person when the CNCP did not cause disability. This figure rose to more than 13,000 USD when the level of interference was severe [86]. The PAIN_Integral Scale© instrument makes it possible to identify the level of impact of CNCP at an early stage, favouring the approach and reducing the health cost, considering the differences between women and men. The use of a single instrument that allows for joint assessment of all impacted areas would facilitate its use by health professionals and would serve as a support in their therapeutic plans.

## 5. Conclusions

The structure of the PAIN_Integral Scale© has remained adequate when analysed in women and men. The results have shown the existence of partial invariance in the structure of the PAIN_Integral Scale© instrument for assessing CNCP in daily life between women and men. This fact could explain the differences in the understanding of the eight items due to the particular idiosyncrasies of both groups, where gender stereotypes learned through differential gender socialisation seem to play an important role. Integrating sex and gender analysis into the study of CNCP is necessary to adapt the treatment to women, who have traditionally been treated with treatments made for men.

## Figures and Tables

**Table 1 healthcare-12-00695-t001:** Data collections.

Age: 16–44 years old, 45–64 years old and over 65 years old. Centre: Virgen del Rocio University Hospital; Virgen Macarena University Hospital; Valme University Hospital; and San Juan de Dios Aljarafe Hospital and Primary Healthcare Centres. Marital status: married, unmarried, separated/divorced and widowed. Employment situation: employed, unemployed, retired/medical leave, homemaker and student. Level of education: early childhood education, primary school, secondary school and higher education. Type of town: fewer than 10,000 inhabitants, from 10,000 to 50,000 inhabitants, more than 50,000 inhabitants and capitals. Location of chronic pain: cervical spine; thoracic spine; lumbar spine; sacral bone; shoulder; armpit/side/arm; elbow; wrist/hand; hips; legs; knee; ankle/foot; stomach; abdomen; facial; migraine/headache; and fibromyalgia. PAIN_Integral Scale©: severe impact, moderate impact and mild impact

**Table 2 healthcare-12-00695-t002:** Descriptive analysis. Sociodemographic characteristics of the sample.

Variables (*n* = 400)	Women (*n* = 252)	Men (*n* = 148)	*p*-Value
	Mean (SD)	Mean (SD)	*p* < 0.001 *
	62.0 (14.0)	56.2 (14.0)	
Age	% (SD)	% (SD)	*p* < 0.05 **
16–44	10.6 (3.8)	17.8 (6.2)	
45–64	46.7 (6.2)	56.8 (24.5)	
65+	42.3 (6.1)	24.7 (6.9)	
	Mean (SD)	Mean (SD)	*p* = 0.11 *
	121.47 (15.58)	123.49 (14.14)	
PAIN_Integral Scale©	% (SD)	% (SD)	*p* = 0.82 **
Severe impact (36–130)	68.7 (5.8)	67.2 (7.6)	
Moderate impact (131–135)	13.3 (4.2)	12.3 (5.3)	
Mild impact (136–180)	18.0 (4.8)	20.5 (6.5)	
Centre			*p* < 0.05 **
Virgen del Rocio University Hospital	77.5 (5.2)	74.0 (1.1)	
Virgen Macarena University Hospital	6.4 (1.6)	11.0 (5.1)	
Virgen de Valme University Hospital	3.2 (1.1)	7.5 (2.2)	
San Juan de Dios Aljarafe Hospital	2.4 (1.9)	2.1 (2.1)	
Primary Healthcare Centres	10.5 (3.8)	7.4 (4.2)	
Marital status			*p* < 0.05 **
Married	55.3 (6.5)	76.0 (6.1)	
Unmarried	12.6 (4.1)	12.3 (5.3)	
Separated/Divorced	10.6 (3.8)	9.6 (4.8)	
Widowed	21.1 (5.1)	1.4 (1.4)	
Employment situation			*p* < 0.05 **
Employed	15.4 (4.5)	24.7 (6.9)	
Unemployed	4.9 (2.7)	6.8 (4.1)	
Retired/medical leave	60.2 (6.1)	66.4 (8.3)	
Homemaker	18.7 (4.8)	0.7 (0.7)	
Student	0.8 (0.8)	0.7 (0.7)	
Level of education			*p* < 0.05 **
Early childhood education	15.4 (4.5)	5.5 (3.7)	
Primary school	54.9 (6.2)	52.7 (8.1)	
Secondary school	17.1 (4.7)	29.5 (11.0)	
Higher education	12.6 (4.1)	11.0 (5.1)	
Type of town			*p* < 0.05 **
Fewer than 10,000 inhabitants	17.7 (4.7)	20.5 (6.5)	
From 10,000 to 50,000 inhabitants	38.2 (6.0)	38.4 (7.9)	
More than 50,000 inhabitants	1.2 (1.2)	1.4 (1.4)	
Capitals	43.0 (6.1)	39.7 (8.0)	
Location of chronic non-cancer pain			*p* < 0.05 **
Cervical spine	23.2 (5.2)	16.4 (6.0)	
Thoracic spine	12.6 (4.1)	7.5 (4.3)	
Lumbar spine	54.9 (6.2)	61.6 (7.9)	
Sacral bone	25.1 (5.7)	26.6 (7.2)	
Shoulder	18.3 (4.8)	6.8 (4.1)	
Armpit/side/arm	15.9 (4.5)	11.0 (5.1)	
Elbow	7.3 (3.2)	2.7 (2.6)	
Wrist/hand	17.5 (4.7)	8.2 (4.4)	
Hips	15.4 (4.5)	10.3 (4.9)	
Legs	36.6 (6.0)	43.3 (8.0)	
Knee	19.5 (4.9)	15.8 (5.9)	
Ankle/foot	16.7 (4.6)	11.0 (5.1)	
Stomach	3.7 (2.3)	0.0 (0.0)	
Abdomen	6.1 (3.0)	4.8 (3.5)	
Facial	1.2 (1.2)	0.7 (0.7)	
Migraine/headache	10.2 (3.8)	7.5 (4.3)	
Fibromyalgia	27.6 (5.5)	6.2 (3.9)	

* Student’s T-test; ** Chi-square test; SD: Standard deviation.

**Table 3 healthcare-12-00695-t003:** Standardised factor loadings of the PAIN_Integral Scale© by sex.

Items	Mobility W (m)	Self-Care W (m)	Sleep W (m)	Treatment Compliance W (m)	Proactivity W (m)	Resilience W (m)	Support network W (m)	Hopelessness Due to Pain W (m)	Pain Catastrophising W (m)
Item 1. Over the last week, how many days have you visited your family or friends?	0.57 (0.70)								
Item 2. Over the last week, how many days have you gone for a walk?	0.64 (0.73)								
Item 3. Over the last week, how many days have you gone shopping?	0.77 (0.77)								
Item 4. Over the last week, how many days have you gotten dressed alone?		0.86 (0.86)							
Item 5. Over the last week, how many days have you carried out personal hygiene practices alone?		0.97 (0.93)							
Item 6. Over the last week, how many days have you carried out your personal grooming?		0.93 (0.91)							
Item 7. In the last week, how many days have you had difficulty achieving restful sleep?			0.78 (0.79)						
Item 8. In the last week, how many days have you been worried or noticed tiredness or a decrease in your socio-labour functioning due to not having slept well the night before?			0.90 (0.96)						
Item 9. In the last week, how many days have you felt too drowsy, falling asleep during the day or sleeping more than usual at night?			0.62 (0.72)						
Item 10. In the last week, how many days did you forget to take your medication?				0.81 (0.81)					
Item 11. In the last week, how many days did you not take your medication because you felt okay?				0.50 (0.66)					
Item 12. In the last week, how many days did you not take your medication because you felt bad?				0.62 (0.44)					
Item 13. In the last week, how many days did you not take your medication because you didn’t want to take so many drugs?				0.62 (0.78)					
Item 14. In the last week, how many days did you not take your medication because you thought it wouldn’t work?				0.60 (0.51)					
Item 15. I’m trying to get them to explain what I can do to lessen the pain.					0.68 (0.70)				
Item 16. I try to know more about my pain so I can cope.					0.72 (0.79)				
Item 17. I’m talking to someone who can do something specific about my pain.					0.79 (0.56)				
Item 18. I’m able to adapt to changes.						0.61 (0.65)			
Item 19. I can overcome any challenge that is presented to me.						0.72 (0.61)			
Item 20. I think I’m a strong person.						0.66 (0.66)			
Item 21. I can handle unpleasant feelings.						0.57 (0.72)			
Item 22. I’m proud of my accomplishments.						0.69 (0.56)			
Item 23. Do you have someone you can count on when you need to talk?							0.79 (0.81)		
Item 24. Do you have someone to take you to the doctor when you need it?							0.53 (0.71)		
Item 25. Do you have someone to show you love and affection?							0.73 (0.80)		
Item 26. Do you have someone to inform you and help you understand the situation?							0.78 (0.77)		
Item 27. Do you have someone to do things with you to forget your problems?							0.90 (0.90)		
Item 28. Do you have someone to help you with your household chores if you are sick?							0.52 (0.74)		
Item 29. Do you have anyone to have fun with?							0.92 (0.89)		
Item 30. Do you have someone who understands your problems?							0.84 (0.76)		
Item 31. The pain is very strong, and I don’t think it’s ever going to get better.								0.49 (0.61)	
Item 32. The pain is very unpleasant, and I feel like I’m out of it.								0.85 (0.79)	
Item 33. I feel like I can’t stand the pain anymore.								0.87 (0.86)	
Item 34. I feel like I don’t have the strength to fight anymore.									0.78 (0.82)
Item 35. I don’t care what could happen to me anymore.									0.88 (0.86)
Item 36. I feel I have lost my emotional stamina.									0.87 (0.85)

M: men; W: women.

**Table 4 healthcare-12-00695-t004:** Comparison of the fit index between invariance models.

Model	χ^2^	χ^2^/(df)	CFI	RMSEA (95% CI)	χ^2^_diff_	ΔCFI	ΔRMSEA
Unconstrained	1739.90	1.56	0.913	0.038 (0.034–0.041)			
Metric invariance model	1802.61	1.58	0.908	0.038 (0.035–0.042)	62.71 (27) *p* < 0.001	−0.005	0.000
Scalar invariance model	1862.78	1.58	0.905	0.038 (0.035–0.042)	60.17 (36) *p* < 0.001	−0.003	0.000
Strict invariance model	2048.18	1.63	0.900	0.040 (0.037–0.043)	126.76 (36) *p* < 0.001	−0.005	0.002

χ^2^ = Chi Square test; χ^2^_diff_ = χ^2^ difference test; CFI = Comparative Fit Index; CI = Confidence interval; RMSEA = Root-Mean-Squared error of Approximation.

**Table 5 healthcare-12-00695-t005:** Item Differential Functioning.

Item Response	Women % (SD)	Men % (SD)	*p*-Value *
Item 1. Over the last week, how many days have you visited your family or friends?			0.451 ^a^
6–7 days	32.5 (5.8)	37.0 (7.8)	
4–5 days	2.8 (2.07)	4.8 (4.8)	
3 days	8.4 (3.5)	6.8 (4.1)	
1–2 days	19.3 (4.9)	2.9 (1.9)	
No day	36.9 (6.0)	29.6 (7.4)	
Item 2. Over the last week, how many days have you gone for a walk?			0.121 ^a^
6–7 days	44.2 (6.1)	50.7 (8.1)	
4–5 days	2.8 (2.0)	6.8 (4.1)	
3 days	6.0 (3.0)	6.8 (4.1)	
1–2 days	15.7 (4.5)	11.0 (5.0)	
No day	31.3 (5.7)	24.7 (6.9)	
Item 3. Over the last week, how many days have you gone shopping?			0.991 ^a^
6–7 days	81.9 (4.7)	82.2 (6.2)	
4–5 days	1.2 (1.1)	1.4 (1.4)	
3 days	1.6 (0.9)	1.4 (1.4)	
1–2 days	2.0 (1.7)	1.4 (1.4)	
No day	13.3 (4.2)	13.7 (5.5)	
Item 4. Over the last week, how many days have you gotten dressed alone?			0.991 ^a^
6–7 days	81.9 (4.7)	82.2 (6.2)	
4–5 days	1.2 (1.1)	1.4 (1.4)	
3 days	1.6 (1.1)	1.4 (1.4)	
1–2 days	2.0 (1.7)	1.4 (1.4)	
No day	13.3 (4.2)	13.7 (5.5)	
Item 5. Over the last week, how many days have you carried out personal hygiene practices alone?			0.689 ^a^
6–7 days	83.5 (4.6)	83.6 (6.0)	
4–5 days	0.8 (0.5)	2.1 (2.1)	
3 days	1.6 (1.1)	0.7 (1.3)	
1–2 days	2.4 (1.9)	3.4 (2.9)	
No day	11.6 (3.9)	10.3 (4.9)	
Item 6. Over the last week, how many days have you carried out your personal grooming?			0.522 ^a^
6–7 days	83.5 (4.6)	88.4 (5.2)	
4–5 days	1.6 (1.1)	0.7 (1.3)	
3 days	2.0 (1.7)	0.7 (1.3)	
1–2 days	3.2 (2.2)	1.4 (1.4)	
No day	9.6 (3.7)	8.9 (4.6)	
Item 7. In the last week, how many days have you had difficulty achieving restful sleep?			0.359 ^a^
6–7 days	47.4 (6.2)	41.8 (7.9)	
4–5 days	10.4 (3.8)	10.3 (4.9)	
3 days	10.8 (3.8)	7.5 (44.2)	
1–2 days	6.4 (3.0)	10.3 (4.9)	
No day	2.9 (2.1)	30.1 (7.4)	
Item 8. In the last week, how many days have you been worried or noticed tiredness or a decrease in your socio-labour functioning due to not having slept well the night before?			0.210 ^a^
6–7 days	44.2 (6.1)	43.8 (8.0)	
4–5 days	9.2 (3.6)	8.9 (4.6)	
3 days	9.2 (3.6)	6.8 (4.1)	
1–2 days	10.0 (3.7)	4.8 (2.1)	
No day	27.3 (5.5)	35.6 (7.7)	
Item 9. In the last week, how many days have you felt too drowsy, falling asleep during the day or sleeping more than usual at night?			0.652 ^a^
6–7 days	37.8 (6.0)	34.2 (7.6)	
4–5 days	7.2 (3.2)	6.2 (3.9)	
3 days	10.8 (3.8)	8.9 (4.6)	
1–2 days	8.8 (4.1)	7.5 (4.2)	
No day	35.3 (5.9)	43.2 (8.0)	
Item 10. In the last week, how many days did you forget to take your medication?			0.473 ^a^
6–7 days	6.4 (3.0)	10.3 (4.9)	
4–5 days	2.8 (2.0)	2.1 (2.3)	
3 days	1.6 (1.1)	3.4 (2.9)	
1–2 days	8.0 (3.3)	7.5 (4.2)	
No day	81.1 (4.8)	76.7 (6.8)	
Item 11. In the last week, how many days did you not take your medication because you felt okay?			0.522 ^a^
6–7 days	1.2 (1.1)	2.7 (2.6)	
4–5 days	0.8 (0.5)	0.7 (1.3)	
3 days	0.0 (0.0)	0.7 (1.3)	
1–2 days	1.2 (1.1)	0.7 (1.3)	
No day	96.8 (2.2)	95.2 (3.4)	
Item 12. In the last week, how many days did you not take your medication because you felt bad?			0.574 ^a^
6–7 days	2.3 (1.9)	2.7 (2.6)	
4–5 days	0.1 (0.3)	0.7 (1.3)	
3 days	0.0 (0.0)	0.0 (0.0)	
1–2 days	1.2 (1.1)	0.7 (1.3)	
No day	96.4 (2.3)	95.9 (3.2)	
Item 13. In the last week, how many days did you not take your medication because you didn’t want to take so many drugs?			0.352 ^a^
6–7 days	3.2 (2.2)	5.5 (10.9)	
4–5 days	0.8 (0.5)	0.7 (1.3)	
3 days	0.8 (0.5)	0.0 (0.0)	
1–2 days	2.8 (2.0)	0.7 (1.3)	
No day	92.4 (3.3)	93.2 (4.1)	
Item 14. In the last week, how many days did you not take your medication because you thought it wouldn’t work?			0.533 ^a^
6–7 days	2.8 (2.0)	1.4 (1.9)	
4–5 days	0.0 (0.0)	0.7 (1.3)	
3 days	0.4 (0.4)	0.0 (0.0)	
1–2 days	0.8 (0.5)	0.1 (0.5)	
No day	96.8 (2.2)	97.8 (2.4)	
Item 15. I’m trying to get them to explain what I can do to lessen the pain.			0.561 ^a^
Never	24.1 (5.3)	18.5 (6.3)	
Rarely	0.0 (0.0)	0.0 (0.0)	
Sometimes	9.6 (3.7)	11.0 (5.0)	
Most of the time	11.6 (3.9)	14.4 (5.7)	
Always	54.6 (6.1)	56.2 (8.0)	
**Item 16. I try to know more about my pain so I can cope.**			**0.011 ^b^**
**Never**	**45.4 (6.1)**	**30.8 (7.4)**	
**Rarely**	**0.0 (0.0)**	**0.0 (0.0)**	
**Sometimes**	**5.2 (2.7)**	**11.6 (5.2)**	
**Most of the time**	**8.4 (3.5)**	**11.0 (5.0)**	
**Always**	**41.0 (6.1)**	**46.6 (8.0)**	
Item 17. I’m talking to someone who can do something specific about my pain.			0.338 ^a^
Never	33.3 (5.8)	25.3 (7.0)	
Rarely	0.0 (0.0)	0.0 (0.0)	
Sometimes	10.8 (3.8)	14.4 (5.7)	
Most of the time	12.0 (4.0)	11.6 (5.2)	
Always	43.8 (6.1)	48.6 (8.1)	
Item 18. I’m able to adapt to changes.			0.859 ^a^
Never	12.4 (4.1)	15.1 (5.8)	
Rarely	0.0 (0.0)	0.0 (0.0)	
Sometimes	15.7 (11.2)	15.8 (5.9)	
Most of the time	18.1 (4.7)	15.8 (5.9)	
Always	53.8 (6.1)	53.4 (8.0)	
Item 19. I can overcome any challenge that is presented to me.			0.100 ^a^
Never	14.9 (4.4)	24.0 (6.9)	
Rarely	0.0 (0.0)	0.0 (0.0)	
Sometimes	21.3 (5.0)	22.6 (6.7)	
Most of the time	20.5 (5.0)	19.2 (6.3)	
Always	43.4 (6.1)	34.2 (7.6)	
**Item 20. I think I’m a strong person.**			**0.008 ^b^**
**Never**	**9.6 (3.7)**	**6.8 (4.1)**	
**Rarely**	**0.0 (0.0)**	**0.0 (0.0)**	
**Sometimes**	**10.4 (3.8)**	**22.6 (6.7)**	
**Most of the time**	**12.9 (4.1)**	**14.4 (5.7)**	
**Always**	**67.1 (3.8)**	**56.2 (8.0)**	
Item 21. I can handle unpleasant feelings.			0.937 ^a^
Never	16.9 (4.6)	18.5 (6.3)	
Rarely	0.0 (0.0)	0.0 (0.0)	
Sometimes	24.1 (5.3)	21.9 (15.2)	
Most of the time	15.3 (4.4)	16.4 (6.0)	
Always	43.8 (6.1)	43.2 (8.0)	
Item 22. I’m proud of my accomplishments.			0.475 ^a^
Never	8.4 (3.5)	11.0 (5.0)	
Rarely	0.0 (0.0)	0.0 (0.0)	
Sometimes	12.9 (4.1)	15.1 (5.8)	
Most of the time	11.6 (3.9)	7.5 (4.2)	
Always	67.1 (3.8)	66.4 (7.6)	
**Item 23. Do you have someone you can count on when you need to talk?**			**0.043 ^b^**
**Never**	**6.0 (2.9)**	**4.1 (3.3)**	
**Rarely**	**6.8 (3.1)**	**6.8 (4.1)**	
**Sometimes**	**13.7 (4.2)**	**6.8 (4.1)**	
**Most of the time**	**8.4 (3.5)**	**15.1 (5.8)**	
**Always**	**65.1 (5.9)**	**67.1 (7.6)**	
Item 24. Do you have someone to take you to the doctor when you need it?			0.430 ^a^
Never	5.6 (3.2)	6.2 (3.9)	
Rarely	4.0 (2.44)	4.1 (0.8–7.2)	
Sometimes	7.2 (3.2)	3.4 (2.9)	
Most of the time	7.2 (3.2)	11.0 (5.0)	
Always	75.9 (5.3)	75.3 (6.9)	
Item 25. Do you have someone to show you love and affection?			0.303 ^a^
Never	2.8 (2.0)	1.4 (1.9)	
Rarely	4.4 (2.5)	2.1 (2.3)	
Sometimes	9.6 (3.7)	6.2 (3.9)	
Most of the time	7.2 (3.2)	10.3 (4.9)	
Always	75.9 (5.3)	80.1 (5.4)	
**Item 26. Do you have someone to inform you and help you understand the situation?**			**0.034 ^b^**
**Never**	**10.0 (3.7)**	**6.8 (4.1)**	
**Rarely**	**7.6 (3.3)**	**8.2 (4.4)**	
**Sometimes**	**16.1 (4.5)**	**6.2 (3.9)**	
**Most of the time**	**8.0 (3.3)**	**8.9 (4.6)**	
**Always**	**58.2 (6.1)**	**69.9 (7.4)**	
**Item 27. Do you have someone to do things with you to forget your problems?**			**0.019 ^b^**
**Never**	**8.8 (4.1)**	**8.2 (4.4)**	
**Rarely**	**9.6 (3.7)**	**2.7 (2.6)**	
**Sometimes**	**14.9 (4.4)**	**8.9 (4.6)**	
**Most of the time**	**7.6 (3.3)**	**11.6 (5.2)**	
**Always**	**59.0 (6.1)**	**68.5 (7.5)**	
**Item 28. Do you have someone to help you with your household chores if you are sick?**			**0.012 ^b^**
**Never**	**12.9 (4.1)**	**6.8 (4.1)**	
**Rarely**	**6.0 (2.9)**	**2.7 (2.6)**	
**Sometimes**	**6.4 (3.0)**	**2.7 (2.6)**	
**Most of the time**	**7.6 (3.3)**	**4.1 (3.3)**	
**Always**	**67.1 (5.8)**	**83.6 (6.0)**	
**Item 29. Do you have anyone to have fun with?**			**0.041 ^b^**
**Never**	**12.0 (4.0)**	**5.5 (10.9)**	
**Rarely**	**7.2 (3.2)**	**4.8 (3.4)**	
**Sometimes**	**11.6 (3.9)**	**6.8 (4.1)**	
**Most of the time**	**7.2 (3.2)**	**8.9 (4.6)**	
**Always**	**61.8 (6.0)**	**74.0 (7.1)**	
**Item 30. Do you have someone who understands your problems?**			**0.047 ^b^**
**Never**	**8.8 (4.1)**	**6.2 (3.9)**	
**Rarely**	**9.6 (3.7)**	**4.1 (3.3)**	
**Sometimes**	**13.7 (4.2)**	**11.0 (5.0)**	
**Most of the time**	**11.2 (3.9)**	**8.2 (4.4)**	
**Always**	**56.6 (6.1)**	**70.5 (7.3)**	
Item 31. The pain is very strong and I don’t think it’s ever going to get better.			0.900 ^a^
Never	10.0 (3.7)	12.3 (5.3)	
Rarely	5.2 (2.7)	5.5 (10.9)	
Sometimes	24.5 (5.3)	22.6 (6.7)	
Most of the time	16.9 (4.6)	19.2 (6.3)	
Always	43.4 (6.1)	40.4 (7.9)	
Item 32. The pain is very unpleasant and I feel like I’m out of it.			0.299 ^a^
Never	8.4 (3.5)	9.6 (4.7)	
Rarely	3.6 (2.3)	6.2 (3.9)	
Sometimes	30.1 (6.1)	32.2 (7.5)	
Most of the time	22.1 (5.1)	26.0 (7.1)	
Always	35.7 (5.9)	26.0 (7.1)	
Item 33. I feel like I can’t stand the pain anymore.			0.869 ^a^
Never	13.3 (4.2)	15.8 (5.9)	
Rarely	6.8 (3.1)	7.5 (4.2)	
Sometimes	32.5 (5.8)	32.2 (7.5)	
Most of the time	16.9 (4.6)	18.5 (6.3)	
Always	30.5 (5.7)	26.0 (7.1)	
Item 34. I feel like I don’t have the strength to fight anymore.			0.547 ^a^
Never	17.3 (4.7)	17.8 (6.2)	
Rarely	8.4 (3.45)	6.8 (4.1)	
Sometimes	28.5 (5.6)	35.6 (7.7)	
Most of the time	30.9 (5.7)	24.7 (6.9)	
Always	14.9 (4.4)	15.1 (5.8)	
Item 35. I don’t care what could happen to me anymore.			0.127 ^a^
Never	11.2 (3.9)	13.0 (5.4)	
Rarely	27.3 (5.5)	18.5 (6.3)	
Sometimes	19.7 (4.9)	28.8 (7.3)	
Most of the time	6.0 (2.9)	7.5 (4.2)	
Always	35.7 (5.9)	32.2 (7.5)	
Item 36. I feel I have lost my emotional stamina.			0.306 ^a^
Never	10.0 (3.7)	12.3 (5.3)	
Rarely	29.3 (5.6)	19.9 (6.4)	
Sometimes	25.3 (5.4)	26.0 (7.1)	
Most of the time	8.8 (4.1)	11.6 (5.2)	
Always	26.5 (5.4)	30.1 (7.4)	

* Chi-Square test; ^a^ Small effect size (Φ ≤ 30); ^b^ Medium effect size (30 < Φ ≤ 50); Use of bold: Questions with differential item compression.

## Data Availability

Data may be obtained from the corresponding author.

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
