# Peer review of "Understanding the Impact of Chronic Non-Cancer Pain on Daily Life from a Gender Perspective Using the PAIN_Integral Scale©"

_healthcare, 2024, doi:10.3390/healthcare12060695_

Round 1

Reviewer 1 Report

Comments and Suggestions for Authors

The article is quite difficult to read and understand. I think the authors should focus on revising the manuscript so as to make it more interesting for readers. Some additional comments:

The title needs to be revised because it is somewhat unclear. Perhaps replacing Comprehension with Understanding?

The Abstract is quite difficult to understand and should undergo significant revision. The authors should focus on communicating their findings in a clear and concise manner, so as the reader understands the main findings of the study.

The PAIN_Integral Scale should be discussed in more detail. Data about validation, availability in other languages, items should be provided.

The Data Analysis section is very technical and provides a lot of information. Perhaps it can be revised and some of the very technical parts can be moved to the Supplementary Data section.

There are significant differences between groups (male versus female) at baseline in terms of age, marital status, pain location and employment. I think this is a very important issue, since these variables can significantly influence pain perception. Did the authors address this bias?

The results section offers a lot of technical details about how results were obtained (some of which was already described in the Material and Method section), but does not truly explain what these results are, despite the long tables detailing participant responses. Also, despite reading the article twice I did not understand if the authors think that the gender differences are due to different understanding of the questionnaire items or due to differences in pain perception and social environment that can be encountered across genders.

Comments on the Quality of English Language

Significant English revisions are required. Some (not all) examples are provided below:

“The experience of Chronic Non-Cancer Pain is uneven between women and men due to differential gender socialisation.”;  “Invariant analysis was performed and to assess whether any of the items on the instrument show a different behaviour in the construct measured by groups Item Differential Functioning was evaluated.”; “Consensus exits that a correct biopsychosocial assessment centred on the person improves the intensity of CNCP (Ministry of Health, Social Services and Equality, 2014), but fundamentally in relation to its impact on daily life (Ministry of Health, Social Services and Equality, 2014; Sánchez-Jiménez et al., 2015)”; “gender stereotypes resulting from the interaction of sex with differential gender socialisation in the experience of CNCP throughout life (Mogil, 2020).”; “Results from the MGCFA and χ2diff test indicated that moderation on sex was present at the model level.”;

Author Response

Dear Editors and Reviewers,

Thank you for reviewing the manuscript and for your comments. Please find below in the text the response to each comment. All the changes in the manuscript have been modified with the marked/tracked version

First Reviewer, Report 1

Quality of English Language

( ) I am not qualified to assess the quality of English in this paper
( ) English very difficult to understand/incomprehensible
(x) Extensive editing of English language required
( ) Moderate editing of English language required
( ) Minor editing of English language required
( ) English language fine. No issues detected

Yes

Can be improved

Must be improved

Not applicable

Does the introduction provide sufficient background and include all relevant references?

( )

( )

(x)

( )

Are all the cited references relevant to the research?

(x)

( )

( )

( )

Is the research design appropriate?

(x)

( )

( )

( )

Are the methods adequately described?

( )

( )

(x)

( )

Are the results clearly presented?

( )

( )

(x)

( )

Are the conclusions supported by the results?

( )

( )

(x)

( )

The article is quite difficult to read and understand. I think the authors should focus on revising the manuscript so as to make it more interesting for readers. Some additional comments:

  1. Muchas gracias por su comentario. El manuscrito ha sido revisado en su totalidad para agilizar y mejorar la lectura y comprensión.

Thank you very much for your comment. The manuscript has been thoroughly reviewed to streamline and enhance readability and comprehension.

  1. The title needs to be revised because it is somewhat unclear. Perhaps replacing Comprehension with Understanding?

Thank you very much for your comment. Following your recommendation, comprehension has been changed to understanding.

  1. The Abstract is quite difficult to understand and should undergo significant revision. The authors should focus on communicating their findings in a clear and concise manner, so as the reader understands the main findings of the study.

Thank you very much for your comment regarding the abstract. Following your recommendation and good judgment, it has been modified, as you can see in the manuscript, and is detailed below:

The experience of chronic non-cancer pain differs between women and men due to gender-related factors. The aim of this study is to assess the difference in responses to the impact of chronic non-cancer pain on daily life in women and men using the PAIN_Integral Scale© instrument. This will be done by means of an analysis of invariance through a multigroup confirmatory factor analysis. A cross-sectional sample of 400 participants over 18 years of age with Chronic Non-Oncological Pain in Pain Units and Primary Care Centres belonging to the Spanish Public Health System was recruited (January to March 2020). An analysis was performed to assess whether any of the items in the instrument showed different behaviour. All analyses were performed using AMOS® v.26 software. The results showed that the structure of the PAIN_Integral© Scale remained adequate when analysing its invariance in women and men, showing no metric, scalar and/or strict invariance. This means that the PAIN_Integral Scale© instrument has a different interpretation for women and men, identifying eight items with a singular functioning in both sexes, and belonging to the subscales of proactivity, resilience and support network. These findings can be explained by gender stereotypes, since the dimensions where there are differences have an important social burden.

  1. The PAIN_Integral Scale should be discussed in more detail. Data about validation, availability in other languages, items should be provided.

Thank you very much for your comment. As you can see, in the section "Instrument with validity and reliability" the main information has been detailed. However, additional information about the instrument has been included. This scale has been validated in Spanish and is currently undergoing validation in Chinese. The items are detailed in Tables 3 and 5.

I attach the text that validates the argument.

“The impact of CNCP on daily life was assessed using the PAIN_Integral Scale©. This instrument facilitates the consolidation of various areas affected by CNCP into a unified scale, expediting the assessment process and considering aspects not addressed by other instruments. The PAIN_Integral Scale© is an instrument composed of 36 items, with a Type-Likert scale from one to five points available in Spanish-language. The results of its validation study showed an adequate reliability (α=0.72) and a structure of nine dimensions that explained 68.22% of the variance. The nine dimensions are self-care, mobility, sleep, treatment compliance, proactivity, resilience, support network, hopelessness due to pain and pain catas-trophising. The results of the Confirmatory Factor Analysis (CFA) demonstrated a good fit for the proposed 9-subscale model Goodness of Fit Index (GFI=0.93), on the Standardised Root Mean Residuals (SRMR=0.05), and on the Root Mean Square Error of Approximation (RMSEA=0.06). Scores on the scale range from 36-180 points that are divided in three in-tervals (36-130: Severe impact; 131-135: Moderate impact; 136-180: Mild impact) (Cáce-res-Matos & Gil-García et al., 2021; Cáceres-Matos & Gil-García et al., 2023).”

  1. The Data Analysis section is very technical and provides a lot of information. Perhaps it can be revised and some of the very technical parts can be moved to the Supplementary Data section.

Thank you very much for your comment. We have included so much technical data to allow readers to reproduce these analyses at other scales. However, we have simplified some of the content.

  1. There are significant differences between groups (male versus female) at baseline in terms of age, marital status, pain location and employment. I think this is a very important issue, since these variables can significantly influence pain perception. Did the authors address this bias?

Thank you very much for your comment. A section on these variables has been included in the limitations section.

I attach the text that validates the argument.

“Despite the differences found, we must take into account certain variables that can modify the response, mainly marital status and age. Marital status may be related to the participants' support network. On the other hand, age is intimately related to the generational experiences where the individual develops and therefore this may influence gender roles.”

  1. The results section offers a lot of technical details about how results were obtained (some of which was already described in the Material and Method section), but does not truly explain what these results are, despite the long tables detailing participant responses. Also, despite reading the article twice I did not understand if the authors think that the gender differences are due to different understanding of the questionnaire items or due to differences in pain perception and social environment that can be encountered across genders.

Dear reviewer, thank you very much for such a pertinent comment. The results section has been clarified to make it more understandable. Regarding your concern, the analyses have been conducted disaggregated by gender, which is what the invariance analyses measure. Once the items with differential understanding between women and men were identified, an attempt has been made to find an explanation from a gender perspective. That is, this involves examining how differential gender socialization due to cultural and, of course, social factors influences how women and men experience life. In this sense, CNCP has a significant subjective component, just like the impact of CNCP on daily life, and therefore, this subjectivity is heavily influenced by sociocultural factors that shape the roles women and men play. Therefore, what we aim to do is identify the items influenced by this differential socialization. Given that you did not clearly understand our intention, it means we were not able to convey our idea in a straightforward manner to the reader. Therefore, we have clarified the discussion. Thank you again.

  1. Comments on the Quality of English Language

Significant English revisions are required. Some (not all) examples are provided below:

“The experience of Chronic Non-Cancer Pain is uneven between women and men due to differential gender socialisation.”;  “Invariant analysis was performed and to assess whether any of the items on the instrument show a different behaviour in the construct measured by groups Item Differential Functioning was evaluated.”; “Consensus exits that a correct biopsychosocial assessment centred on the person improves the intensity of CNCP (Ministry of Health, Social Services and Equality, 2014), but fundamentally in relation to its impact on daily life (Ministry of Health, Social Services and Equality, 2014; Sánchez-Jiménez et al., 2015)”; “gender stereotypes resulting from the interaction of sex with differential gender socialisation in the experience of CNCP throughout life (Mogil, 2020).”; “Results from the MGCFA and χ2diff test indicated that moderation on sex was present at the model level.”;

Many thanks for your appreciation of the English revision. The manuscript was revised by a professional translator who is a native English speaker. However, it has been rechecked for flaws in the wording.

Reviewer 2 Report

Comments and Suggestions for Authors

This article reports data from a questionnaire that measures pain impact and tests whether responses are consistent between male and female respondents. It outlines differences and attempts to consider possible reasons for these differences in relation to other research.

While some differences are identified, the consideration of these on an individual basis does not meet the stated outcome of the research of detecting differences between men and women that might influence clinical treatment. In this aim I believe that the authors have overstated their findings.

Introduction line 67 to 69 It is stated that the impact of pain is known to differ according to cultural factors and socioeconomic status. This is true, however it does not appear to be relevant to the research as the article does not explore differences according to any factors other than sex.

The authors state that their power calculation indicated that 137 subjects would be needed in order to reach significance if there was a large effect size between groups. The inclusion of more than 400 subjects therefore means that there is a risk that significant findings will be found with a far smaller and possibly unimportant effect size which seriously limits the reliability of the findings.

The details summarised from lines 129 to 135 are not necessary for the current study and include a number of self-citations by the study authors.

The information in lines 145 to 155 is difficult to read and would be better presented in a table.

I am unable to comment on the invariant analysis method as it is not a method that I am familiar with.

Line 198 appears to include writing in the future tense which has perhaps not been changed from the original protocol.

Consent is described as “written or verbal”. More details about the collection of verbal consent would be necessary in order to confirm that adequate ethical process has been followed.

I am not clear what the p values in table 1 refer to. Do they indicate differences in the numbers of women and men in the study for example, or differences in the score achieved between women and men. If the former then this detail is not helpful or needed.

Although this paper is not primarily about the scale in question it should be noted that the questions may not measure the impact of pain. For example, the first three articles that refer to mobility ask whether a person has for example gone for a walk, but does not ask whether they have been prevented by pain from going for a walk. It does not therefore distinguish between activities limited by pain and activities not undertaken for different reasons. This needs to be identified as a limitation.

Looking at the item differences between men and women some seem to be quite small from the raw data, which is important given the large sample size and risk of significance attributed to small differences. A more robust method would be to split the sample randomly in half and carry out statistics on both halfs of the sample. This would result in sample sizes closer to that calculated and also provide quality assurance if significant results were obtained for both halfs of the dataset.

Line 272 the authors state that the results indicate that women and men show a different understanding of the scale. An alternative possibility would be that each construct actually does differ between women and men – for example women genuinely do feel stronger, are stronger compared to men. The discussion should consider both possibilities.

Again, I am wary of the discussion of individual items and possible reasons for differences given the statistical weaknesses outlined above.

It might have been interesting to outline where there were similarities between the responses of women and men as well as where there were differences – such an approach may add insight to the discussion.

Comments on the Quality of English Language

English is overall good with just a couple of minor errors

Author Response

Second Reviewer, Report 1

Quality of English Language

( ) I am not qualified to assess the quality of English in this paper
( ) English very difficult to understand/incomprehensible
( ) Extensive editing of English language required
( ) Moderate editing of English language required
(x) Minor editing of English language required
( ) English language fine. No issues detected

Yes

Can be improved

Must be improved

Not applicable

Does the introduction provide sufficient background and include all relevant references?

( )

(x)

( )

( )

Are all the cited references relevant to the research?

( )

( )

(x)

( )

Is the research design appropriate?

( )

(x)

( )

( )

Are the methods adequately described?

( )

( )

( )

(x)

Are the results clearly presented?

( )

( )

(x)

( )

Are the conclusions supported by the results?

( )

( )

(x)

( )

  1. This article reports data from a questionnaire that measures pain impact and tests whether responses are consistent between male and female respondents. It outlines differences and attempts to consider possible reasons for these differences in relation to other research. While some differences are identified, the consideration of these on an individual basis does not meet the stated outcome of the research of detecting differences between men and women that might influence clinical treatment. In this aim I believe that the authors have overstated their findings.

The aim of this study was not to establish differences between men and women to impact clinical treatment. Our objective was, as indicated in the introduction: 'The objectives of this study were to test the structure of the PAIN_Integral Scale© using MGCFA to assess its invariance in women and men and to analyze the items that show a differential functioning between both groups.' While it is true, as we also mentioned in the introduction and discussion sections, that differences in how women and men perceive how CNCP impacts daily life can influence the assessment of clinical aspects, such as the self-perception of impact. In this sense, these experiences influence clinical practice, and therefore the treatment that women and men seek. However, the manuscript has been reviewed with the intention of clarifying this aspect and preventing these types of confusions.

  1. Introduction line 67 to 69 It is stated that the impact of pain is known to differ according to cultural factors and socioeconomic status. This is true, however it does not appear to be relevant to the research as the article does not explore differences according to any factors other than sex.

Thank you very much for your comment. Only sex has been valued since it has an intimate relationship with the social construct known as gender. However, introduction section has been modified to be clearer.

  1. The authors state that their power calculation indicated that 137 subjects would be needed in order to reach significance if there was a large effect size between groups. The inclusion of more than 400 subjects therefore means that there is a risk that significant findings will be found with a far smaller and possibly unimportant effect size which seriously limits the reliability of the findings.

Thank you very much for your comment. The error has been corrected. The parameters of the tests used are attached. Therefore the total sample corresponds to 386 participants.

  1. The details summarised from lines 129 to 135 are not necessary for the current study and include a number of self-citations by the study authors.

Thank you very much for your appreciation and we understand the concern of the scientific community about the existence of self-citations. In this case we have decided to keep the section and the citations, since we consider that it is necessary to specify the information regarding the psychometric properties of the validation study of the original instrument. Far from being eager to self-cite our own studies, we intend to include information that we consider necessary. This scale has been designed to bring together different dimensions related to pain. It has currently aroused the interest of Chinese-speaking scientists and is currently being validated in Chinese.

  1. The information in lines 145 to 155 is difficult to read and would be better presented in a table.

Thank you very much for your recommendation. We really feel that it improves the clarity of the presentation of the data and following your recommendations we have modified it.

  1. I am unable to comment on the invariant analysis method as it is not a method that I am familiar with.
  2. Line 198 appears to include writing in the future tense which has perhaps not been changed from the original protocol.

Thank you very much for your appreciation. It is a mistake in the wording of the sentence.

  1. Consent is described as “written or verbal”. More details about the collection of verbal consent would be necessary in order to confirm that adequate ethical process has been followed.

Dear reviewer, we understand your concern about whether all ethical aspects have been properly respected and we assure you that this has been done. Given that the data collection for the study did not involve a modification of the treatment, evolution or course of the disease, we understand that this is a low-risk study for the participants. Therefore, with the permission of the Research Ethics Committee, verbal consent was allowed from the participants. As with those who gave written consent, they were given the participant information sheet and all aspects of the study were explained to them, inviting them to participate. The participants responded affirmatively to their interest in taking part in the study. In view of possible confusion, this information has been included and clarified in the manuscript.

  1. I am not clear what the p values in table 1 refer to. Do they indicate differences in the numbers of women and men in the study for example, or differences in the score achieved between women and men. If the former then this detail is not helpful or needed.

The p-values indicate the difference between men and women with the different sociodemographic variables collected in the study. This allows us to understand the initial situation of the participants and describe the sample. We are aware that it is not necessary or useful, as previously mentioned. However, this practice is an emerging technique in recent studies, and given that one of the members of the research team holds a PhD in Statistical Sciences, we decided to include these calculations under their judgment and advice.

  1. Although this paper is not primarily about the scale in question it should be noted that the questions may not measure the impact of pain. For example, the first three articles that refer to mobility ask whether a person has for example gone for a walk, but does not ask whether they have been prevented by pain from going for a walk. It does not therefore distinguish between activities limited by pain and activities not undertaken for different reasons. This needs to be identified as a limitation.

Thank you very much for your comment. The conception of this instrument is to evaluate the impact of CNCP on daily life, irrespective of its etiological origin. As you are aware, this health issue affects numerous aspects of daily life, leading to impairment and limitation. One of the areas it significantly affects is mobility, resulting in challenges with basic activities of daily living. This is the rationale behind our team's decision to include this construct in the preliminary validation study. As you can observe in the mentioned study (https://onlinelibrary.wiley.com/doi/10.1111/jan.14877), this subscale demonstrated a Cronbach's alpha of 0.75, signifying a robust score.

  1. Looking at the item differences between men and women some seem to be quite small from the raw data, which is important given the large sample size and risk of significance attributed to small differences. A more robust method would be to split the sample randomly in half and carry out statistics on both halfs of the sample. This would result in sample sizes closer to that calculated and also provide quality assurance if significant results were obtained for both halfs of the dataset.

This has been corrected in the sample size calculation section. Thank you very much for your comment, as it has allowed us to identify the error.

  1. Line 272 the authors state that the results indicate that women and men show a different understanding of the scale. An alternative possibility would be that each construct actually does differ between women and men – for example women genuinely do feel stronger, are stronger compared to men. The discussion should consider both possibilities.

Thank you very much for your feedback. This has been clarified in the manuscript.

  1. Again, I am wary of the discussion of individual items and possible reasons for differences given the statistical weaknesses outlined above.

This has been corrected in the sample size calculation section. Thank you very much for your comment.

  1. It might have been interesting to outline where there were similarities between the responses of women and men as well as where there were differences – such an approach may add insight to the discussion.

Thank you very much for your insight. We find this aspect to be highly relevant, even though it wasn't one of the study's objectives. Our intention was solely to identify the structure. However, we believe it could be an important prospective avenue, so we have included it in this section of the manuscript.

  1. Comments on the Quality of English Language: English is overall good with just a couple of minor errors

Many thanks for your appreciation of the English revision. The manuscript was revised by a professional translator who is a native English speaker. However, it has been rechecked for flaws in the wording.

Reviewer 3 Report

Comments and Suggestions for Authors

This is a review of the manuscript entitled, "Understanding of the impact of Chronic Non-Cancer Pain on daily life from a gender perspective using the PAIN_Integral Scale©". 

The authors describe a study in which the authors are trying to determine the difference between men and women and their experience of chronic, non cancer pain as measured by the PAIN_Integral Scale. The scale is a measure of overall functioning and quality of life. The authors appear to be applying this scale to a novel subject group. They are also attempting to demonstrate its utility within busy clinic practice. The manuscript may be enhanced by attention to the following considerations listed below.

Title: appears to describe the nature of the study accurately

Introduction: close attention to grammar, written English and syntax may be useful and beneficial for the entire manuscript. For example, P2, lines 48 – 50, the sentence “In addition, the experience of CNCP is uneven in different societies given the connotations that it has in each culture [12,13] and social aspects [14], as well as by sex and gender [15].” Is not entirely clear. It appears the authors are trying to state that sex and gender are one social aspect of the pain experience that may influence differences between genders. Please clarify this sentence and please check for similar sentences throughout the introduction here

Results:

Table 2: this was somewhat hard to follow as the alignment of the lines were off

Table 3: It may be useful to combine elements of Table 3 and Table 5. Perhaps one could denote the items that had significant differences between the genders by highlighting the items (either bold or with notations). This could significantly save space for the manuscript. Information in Table 5 may not be entirely necessary, only those items that are different should be highlighted.

There are many comparisons and with multiple comparisons in Table 5, it seems there may need to be some corrections for all the multiple comparisons. Please address or provide rationale for why a correction is not necessary.

Discussion: The authors may want to exercise some caution when trying to explain the reasons for the differences on this scale. Without direct input through additional instruments, or asking participants to explain in detail why a response to an item was selected, the speculation listed in lines 274-293 are a bit untethered to any evidence in the current study. Unless all of the gendered females have gone through childbirth or the men subscribe to the THMM model, these are only speculations. Addtionally, the authors’ own statements, line 274 “…resilience construct and showed better scores in women than in men” could be rephrased in terms of direction of scores, not value judgments of “better or worse”. See also statement line 303, “better scores in men…”

Overall, this manuscript adds to the growing literature demonstrating that individuals with gendered identy may sometimes answer items on scales differently. Precisely why the differences do appear remains uncertain. The authors should be commended for exploring this intriguing line of inquiry.

Comments on the Quality of English Language

Minor revisions for the syntax and grammar should be considered. Please see review for specific examples.

Author Response

Reviewer 1

Open Review

Quality of English Language

( ) I am not qualified to assess the quality of English in this paper
( ) English very difficult to understand/incomprehensible
( ) Extensive editing of English language required
(x) Moderate editing of English language required
( ) Minor editing of English language required
( ) English language fine. No issues detected

Yes

Can be improved

Must be improved

Not applicable

Does the introduction provide sufficient background and include all relevant references?

( )

(x)

( )

( )

Are all the cited references relevant to the research?

(x)

( )

( )

( )

Is the research design appropriate?

(x)

( )

( )

( )

Are the methods adequately described?

(x)

( )

( )

( )

Are the results clearly presented?

(x)

( )

( )

( )

Are the conclusions supported by the results?

( )

( )

(x)

( )

Minor revisions for the syntax and grammar should be considered. Please see review for specific examples.

This is a review of the manuscript entitled, "Understanding of the impact of Chronic Non-Cancer Pain on daily life from a gender perspective using the PAIN_Integral Scale©". 

The authors describe a study in which the authors are trying to determine the difference between men and women and their experience of chronic, non cancer pain as measured by the PAIN_Integral Scale. The scale is a measure of overall functioning and quality of life. The authors appear to be applying this scale to a novel subject group. They are also attempting to demonstrate its utility within busy clinic practice. The manuscript may be enhanced by attention to the following considerations listed below.

 Title: appears to describe the nature of the study accurately

  1. Introduction: close attention to grammar, written English and syntax may be useful and beneficial for the entire manuscript. For example, P2, lines 48 – 50, the sentence “In addition, the experience of CNCP is uneven in different societies given the connotations that it has in each culture [12,13] and social aspects [14], as well as by sex and gender [15].” Is not entirely clear. It appears the authors are trying to state that sex and gender are one social aspect of the pain experience that may influence differences between genders. Please clarify this sentence and please check for similar sentences throughout the introduction here

Dear reviewer, thank you very much for your comment. We have attempted to clarify the text to make it more understandable to the reader.

  1. Results:

     4.A. Table 2: this was somewhat hard to follow as the alignment of the lines were off

Dear reviewer, This is due to a formatting error. Thank you very much for your attention to detail. This discrepancy has been corrected in Table 2 to enhance readability.

   4.B. Table 3: It may be useful to combine elements of Table 3 and Table 5. Perhaps one could denote the items that had significant differences between the genders by highlighting the items (either bold or with notations).

This could significantly save space for the manuscript. Information in Table 5 may not be entirely necessary, only those items that are different should be highlighted.

There are many comparisons and with multiple comparisons in Table 5, it seems there may need to be some corrections for all the multiple comparisons. Please address or provide rationale for why a correction is not necessary.

Dear reviewer, We acknowledge your concern regarding the length of Table 5 and your proposal to merge it with Table 3. However, we believe it would be preferable to keep them in separate formats as Table 3 is based on confirmatory factor analysis, which is part of the methodology followed. Table 5 aims to establish differences between different items according to women and men to identify those items that show differential behavior.

Discussion: The authors may want to exercise some caution when trying to explain the reasons for the differences on this scale. Without direct input through additional instruments, or asking participants to explain in detail why a response to an item was selected, the speculation listed in lines 274-293 are a bit untethered to any evidence in the current study. Unless all of the gendered females have gone through childbirth or the men subscribe to the THMM model, these are only speculations. Addtionally, the authors’ own statements, line 274 “…resilience construct and showed better scores in women than in men” could be rephrased in terms of direction of scores, not value judgments of “better or worse”. See also statement line 303, “better scores in men…”

Dear reviewer, we agree with your assessment and have reviewed and modified the manuscript taking into account your considerations.

Overall, this manuscript adds to the growing literature demonstrating that individuals with gendered identy may sometimes answer items on scales differently. Precisely why the differences do appear remains uncertain. The authors should be commended for exploring this intriguing line of inquiry.

Thank you very much for your kind words and for taking the time and effort to review this manuscript, providing us with improvement suggestions that we believe have enhanced the quality of the article.

Reviewer 2

Open Review

Quality of English Language

( ) I am not qualified to assess the quality of English in this paper
( ) English very difficult to understand/incomprehensible
( ) Extensive editing of English language required
(x) Moderate editing of English language required
( ) Minor editing of English language required
( ) English language fine. No issues detected

Yes

Can be improved

Must be improved

Not applicable

Does the introduction provide sufficient background and include all relevant references?

( )

( )

(x)

( )

Are all the cited references relevant to the research?

(x)

( )

( )

( )

Is the research design appropriate?

(x)

( )

( )

( )

Are the methods adequately described?

( )

( )

(x)

( )

Are the results clearly presented?

( )

( )

(x)

( )

Are the conclusions supported by the results?

( )

(x)

( )

( )

Comments and Suggestions for Authors

My comments on the quality of the English language are included in the review report attached above. 

Peer review report: Understanding of the impact of Chronic Non-Cancer Pain on daily life from a gender perspective using the PAIN_Integral 3 Scale©

The manuscript described the study applying the PAIN_Integral 3 Scale© coupled with Multigroup Confirmatory Factor analysis to evaluate the difference between men and women in their response to chronic non-cancer pain in daily life. The topic is relevant to the Journal’s reader. The study is comprehensive, yet the authors organized the  manuscript step-by-step in a structure that is an easy path for the lay audience.

However, the English language used was lengthy and confusing. I present several comments to improve the paper and hope the authors will consider them.

1• Lines 20-23: I suggest the authors combine these two sentences. For example, “This study (1) assessed the difference in responses to the impact of chronic non-cancer pain on daily life in women and men using the PAIN_Integral Scale© and (2) evaluated its invariance through multigroup confirmatory factor analysis.”

R1=Thank you very much for your comments. We have made the suggested changes

2• Line 29: What does “This” refer to? The results? I observed an overarching issue of the use of “this,” “these,” etc., which may be unclear as to what they refer to. I suggest the authors rewrite/combine the sentences, removing the unclear references.

R2= Thank you very much for your comments. We have rewritten that sentence.

3• Lines 63-68: The meaning of this paragraph is unclear. I suggest the authors

re-write using the list to present the reasons mentioned.

R3= Thank you very much for your comment. The sentence has been rewritten for better understanding.

4• Lines 69-74: I suggest the authors either combine this paragraph with the previous one or clearly state what “these differences” are.

R4= Thank you very much for your comment. The sentence has been rewritten for better understanding.

5• Lines 75-80: Same as the above! I suggest the authors either combine this paragraph with the previous one(s) or clearly state what “this technique” is.

R5=Thank you very much for your appreciation. This has been modified in the manuscript

  • Lines 81-83: Sex and gender are often used interchangeably but are two different terms. The authors clarified the difference between the role of sex and gender in pain response above. I assume the study was on gender and suggest the authors state so.

Dear reviewer, Indeed, we have clarified the differences between sex and gender and the implications of each with pain. What we have aimed to do is to analyze the understanding of the impact of non-oncological chronic pain between women and men. On the one hand, as we have indicated in the manuscript, these differences may be due to biological factors that make the experience of pain different, but also to gender, which is influenced by differential gender socialization. Therefore, as you rightly point out, we aim to identify those differences that may be explained by gender.

6• Lines 86-89: The sentences are not flowing smoothly; suggest a rewrite.

R6=This has been clarified in the manuscript according to your suggestions

7• Line 89: What is the “STROBE” checklist?

R7= The "STROBE checklist" is a tool used in research to assess the quality of observational studies in epidemiology. STROBE stands for "Strengthening the Reporting of Observational Studies in Epidemiology." The STROBE checklist provides a detailed guide for authors on what information should be included in reports of observational studies, such as cohort, case-control, and cross-sectional studies. The purpose of this checklist is to improve the transparency and quality of the presentation of results from observational epidemiological research. Given that the study design is cross-sectional, this checklist has been used to assess the quality of this study.

8• Lines 91-92: The sentence is long and ambiguous; suggest breaking it into two. Were McCallum and the COSMIN Checklist Statement recommended the same idea?

R8=This has been clarified in the manuscript

9• Line 94: suggest adding the reference (the website used) for G*Power V.3.1.9.4.

R9= Dear reviewer, The G*Power software is not available online; instead, it has been a version installed on the computer. However, you can find it available for download at the following web address: Universität Düsseldorf: G*Power (hhu.de)

10• Line 118: suggest separating the Data Collection and Data Analysis into two

Subsections.

R10= Dear reviewer, we have made the change you suggested in the manuscript.

11• Table 1: Gender was also collected. Should it be included? I suggest changing Table 1's title to Data Collections, including gender and how the PAIN_Integral Scale variable is categorized.

R11=Dear reviewer, unfortunately, the gender each participant identified with was not collected; only data regarding sex was gathered. Your recommendations regarding Table 1 have been taken into consideration and modified in the manuscript.

12• Line 129: Suggest adding the subsection for Data Analysis and deleting the

phrase “Regarding data analysis.”

R12= Dear reviewer, we have made the change you suggested in the manuscript.

13• Lines 132-134: I suggest consistently using categorial (qualitative) and

continuous (quantitative) variables.

R13=This has been modified in the manuscript

14• Lines 141-142: Sex or gender?

R14=Dear reviewer, here, the term sex is correct because we are separating between women and men without a gender explanation

15• Lines 172-183: Should Ethical considerations be included at the end of the

manuscript (Institutional Review Board Statement)?

R15= Dear reviewer, we have deliberately included the ethical aspects in that section.

16• Line 192: “Sex”? The manuscript title indicated “gender”.

R16=This has been modified in the manuscript.

17• Lines 192-193: I do not understand the sentence.

R17=This has been clarified in the manuscript.

18• Line 193: How was “the nine-solution model” defined?

R18=El modelo de nueve soluciones fue definido, tal y como usted puede comprobar  a partir de los análisis factoriales llevados a cabo en dicho estudio y en el estudio de validación publicado anteriormente en la revista Journal of Advanced Nursing.

19• Line 224: How were “eight items” identified? Statistically significant in the

cross-tab table (Table 5)?

R19= Dear reviewer, as indicated in the methodology section, the differential behaviour of the item was assessed. Since the necessary requirements for the application of the Rasch method were not met, the Chi-Square test was used. This is intended to determine the relationship between the sex variable and the item scores through statistical significance and effect size.

20• Lines 327-333: Self-report data and cross-sectional design also pose some

limitations. Please include them.

R20= They have been included in the limitations section. Thank you for your suggestion

21• Lines 364-372: This paragraph does seem to belong to the introduction to the

background to serve as a rationale for the study rather than as an implication for practice.

R21=Dear reviewer, we understand your perspective, but we have decided not to include that content in the discussion as you recommend since the economic cost associated with pain is not one of the main aspects of the study. However, this study enables us to enhance and reform the utility of using the PAIN_Integral Scale instrument, and therefore, we believe it has relevance and impact on clinical practice.

22• Lines 374-381: Sex or gender or both? Perhaps the paragraphs on lines 51-59

should be revised.

R22=This has been reviewed in the manuscript following your suggestions

Thanks for the opportunity to read well-worthy research

Thank you very much for your kind words and for taking the time and effort to review this manuscript, providing us with improvement suggestions that we believe have enhanced the quality of the article.

Reviewer 4 Report

Comments and Suggestions for Authors

Comments on the Quality of English Language

My comments on the quality of the English language are included in the review report attached above. 

Author Response

(The authors gave the same response as above.)

Round 2

Reviewer 2 Report

Comments and Suggestions for Authors

The abstract contains some future tense which should be changed to past.

The paragraph starting line 266 would better say that differences in responses between men and women were detected, which may reflect differences in interpretation of the questions or alternatively differences in the impact of pain.

Author Response

Reviewer 2

The abstract contains some future tense which should be changed to past.

The paragraph starting line 266 would better say that differences in responses between men and women were detected, which may reflect differences in interpretation of the questions or alternatively differences in the impact of pain.

Dear reviewer, thank you very much for your improvement comments. We consider your suggestions highly appropriate. These have been incorporated into the manuscript.